

# Description of a new member of the family *Erysipelotrichaceae*: *Dakotella fusiforme* gen. nov., sp. nov., isolated from healthy human feces

Sudeep Ghimire, Supapit Wongkuna and Joy Scaria

Department of Veterinary and Biomedical Sciences, South Dakota State University, Brookings, SD, United States of America

## ABSTRACT

A Gram-positive, non-motile, rod-shaped facultative anaerobic bacterial strain SG502[T] was isolated from healthy human fecal samples in Brookings, SD, USA. The comparison of the 16S rRNA gene placed the strain within the family *Erysipelotrichaceae*. Within this family, *Clostridium innocuum* ATCC 14501[T], *Longicatena caecimuris* strain PG-426-CC-2, *Eubacterium dolichum* DSM 3991[T] and *E. tortuosum* DSM 3987[T] (=ATCC 25548[T]) were its closest taxa with 95.28%, 94.17%, 93.25%, and 92.75% 16S rRNA sequence identities respectively. The strain SG502[T] placed itself close to *C. innocuum* in the 16S rRNA phylogeny. The members of genus *Clostridium* within family *Erysipelotrichaceae* was proposed to be reassigned to genus *Erysipelatoclostridium* to resolve the misclassification of genus *Clostridium*. Therefore, *C. innocuum* was also classified into this genus temporarily with the need to reclassify it in the future because of its difference in genomic properties. Similarly, genome sequencing of the strain and comparison with its 16S phylogenetic members and proposed members of the genus *Erysipelatoclostridium*, SG502[T] warranted a separate genus even though its 16S rRNA similarity was >95% when comapred to *C. innocuum*. The strain was 71.8% similar at ANI, 19.8% [17.4–22.2%] at dDDH and 69.65% similar at AAI to its closest neighbor *C. innocuum*. The genome size was nearly 2,683,792 bp with 32.88 mol% G+C content, which is about half the size of *C. innocuum* genome and the G+C content revealed 10 mol% difference. Phenotypically, the optimal growth temperature and pH for the strain SG502[T] were 37 °C and 7.0 respectively. Acetate was the major short-chain fatty acid product of the strain when grown in BHI-M medium. The major cellular fatty acids produced were $C_{18:1} \omega 9c$, $C_{18:0}$ and $C_{16:0}$. Thus, based on the polyphasic analysis, for the type strain SG502[T] (=DSM 107282[T] = CCOS 1889[T]), the name *Dakotella fusiforme* gen. nov., sp. nov., is proposed.

## INTRODUCTION

The members of family *Erysipelotrichaceae* have been isolated from the intestinal tracts of mammals (*Alcaide et al., 2012*; *Greiner & Backhed, 2011*; *Han et al., 2011*) and insects (*Egert et al., 2003*) and are associated with host metabolism and inflammatory diseases

Corresponding author
Joy Scaria, joy.scaria@sdstate.edu, joyscaria@gmail.com

(*Cox et al., 2014*; *Kaakoush, 2015*). Although metagenome analysis of gut microbiome have revealed the composition and function of the microbiome in the intestine, only a few cultured species available from this family and their function in the gut ecosystem is not yet well understood (*Kaakoush, 2015*).

The family *Erysipelotrichaceae* was originally described by *Verbarg et al. (2004)*. The members of this family includes Gram-positive, filamentous rods and were originally described as facultative anaerobes but later amended by *Tegtmeier et al. (2016)* to include obligate anaerobes. The members of this family belonged to *Clostridial* Cluster XVI that consists of three major species; *Clostridium innocuum, Eubacterium biforme* and *Streptococcus pleomorphus* (*Collins et al., 1994*). According to the updated LPSN list of valid bacteria (http://www.bacterio.net/), current members of this family includes the following genera: *Allobaculum, Breznakia, Bulleidia, Catenibacterium, Canteisphaera, Coprobacillus, Dielma, Dubosiella, Eggerthia, Erysipelothrix, Faecalibaculum, Faecalicoccus, Faecalitalea, Holdemania, Kandleria, Longibacculum, Longicatena, Solobacterium,* and *Turicibacter.* Members of this family have low-G+C content and were previously recognized as the ''walled relatives'' of mycoplasma (*Weisburg et al., 1989*) and later classified under *Clostridial* cluster XVI (*Collins et al., 1994*). With major changes in the taxonomy of *Erysipelotrichaceae*, recently some members have been reclassified into new families *Coprobacillaceae* (*Collins et al., 1994*) and *Turicibacteriaceae* (*Verbarg et al., 2014*) but the placement of cluster XVI is still debated. A few species related to *Clostridium* and *Eubacterium* are also included within *Erysipelotrichaceae* based on the 16S rRNA gene sequence similarity and are considered as misclassified (*Verbarg et al., 2014*). For the proper classification of these misclassified members of genus *Clostridium*, different genus names were proposed by Yutin and Galperin in 2013 where *C. innocuum* along with *C. cocleatum, C. saccharogumia* and *C. spiroforme* were proposed to be reassigned to a new genus *Erysipelatoclostridium.* As, *C. innocuum* was found to be more distantly related to other members inside *Erysipelatoclostridum* genus, they also highlighted the need of future reclassification of *C. innocuum* (*Yutin & Galperin, 2013*).

The present study describes the isolation, physiology, and genomic characterization of a new member of the family *Erysipelotrichaceae* isolated from healthy human feces. Within the family *Erysipelotrichaceae*, strain SG502[T] clustered within clostridial cluster XVI. Also, the strain SG502[T] showed less than 97% 16S rRNA gene sequence similarity towards its nearest phylogenetic neighbor *C. innocuum* ATCC 14501[T]. Therefore, we performed in vitro phenotypic characterization and sequenced the genome for comparative analysis with its neighbors. We found major differences in the genomic and phenotypic characteristics of the strain even though its 16S rRNA gene was >95% similar to nearest neighbor *C. innocuum*. Thus, we propose a novel genus and species for this strain and propose designating SG502[T] as *Dakotella fusiforme* gen. nov., sp. nov within the family *Erysipelotrichaceae*.

## MATERIALS AND METHODS

### Bacterial isolation and culture condition

The strain was isolated from healthy human fecal sample as part of a culturomics study. The collection of the human fecal samples were done with the approval of the Institutional

review board (approval #IRB-1709018-EXP) at South Dakota State University, Brookings, SD, USA. The fecal samples were collected after receiving the informed consent form the donors. After transferring the fresh fecal samples into the anaerobic chamber (85% nitrogen, 10% hydrogen and 5% carbon dioxide) within 10 min of voiding, the sample was diluted 10 times with anaerobic PBS and stored with 18% DMSO in -80 °C. The sample was cultured in modified BHI medium (BHI-M) containing 37g/L of BHI, 5 g/L of yeast extract, 1 ml of 1 mg/mL menadione, 0.3 g of L-cysteine, one mL of 0.25 mg/L of resazurin, one mL of 0.5 mg/mL hemin, 10 mL of vitamin and mineral mixture,1.7 mL of 30 mM acetic acid, two mL of 8 mM propionic acid, two mL of 4 mM butyric acid, 100 µl of 1 mM isovaleric acid, and 1% pectin and inulin. After isolation, the strain was subjected to MALDI-ToF (Bruker, Germany). Since MALDI-ToF did not identify a species, 16S rRNA gene sequencing was performed for species identification.

## Phenotypic and chemotaxonomic characterization

For morphological, physiological and biochemical characterization, the strain was cultivated in BHI-M medium in anaerobic conditions at 37 °C at pH $6.8 \pm 0.2$. Colony characteristics were determined after streaking the strain on BHI-M agar plates followed by 48 h of anaerobic incubation. Gram staining was performed using a Gram staining kit (BD Difco) according to the manufacturer's protocol. During the exponential growth of the bacterium, cell morphology and flagellation was examined under scanning electron microscopy (SEM). SG502[T] was grown separately in aerobic and anaerobic conditions to determine the aerotolerance. Further, the strain was grown at 4, 20, 30, 40 and 55 °C to determine the range of growth under anaerobic conditions. The BHI-M media was adjusted to pH levels between 4 and 9 with 0.1N HCl and 0.1N NaOH to determine the growth of the strain at different pH levels. BHI-M medium was supplemented with triphenyltetrazolium chloride (TTC) (*Shields & Cathcart, 2011*) to determine the motility of the strain.

The phenotypic and biochemical characterizations were performed using AN MicroPlate (Biolog) and API ZYM (bioMerieux) according to the manufacturer's instructions. Also, after growing the strain SG502[T] and ATCC 14501[T] in BHI-M medium at 37 °C for 24 h, cells were harvested for cellular fatty acid analysis. Fatty acids were extracted, purified, methylated, identified and analyzed using GC (Agilent 7890A) according to manufacturer's instructions (MIDI) (*Sasser, 1990*). Further, short-chain fatty acid (SCFA) production was determined using gas chromatography after cells were grown in BHI-M medium. For SCFA estimation, 800 µl of the bacterial culture was collected and 160 µl of freshly prepared 25% meta-phosphoric acid (w/v) was added before freezing to $-80$ °C. The sample were thawed and centrifuged at $>20,000 \times g$ for 30 min before injecting 600 µl of the supernatant into the TRACE1310 GC system (ThermoScientific, Waltham, MA, USA).

## Phylogenetic analysis

Genomics DNA from the strain was isolated using E.Z.N.A bacterial DNA isolation kit (Omega Biotek) following the manufacturer's instructions. The 16S rRNA gene was amplified using universal primer set 27F (5′- AGAGTTTGATCMTGGCTCAG-3′) and 1492R (5′- ACCTTGTTACGACTT- 3′) and sequenced using a Sanger sequencing chemistry
(ABI 3730XL; Applied Biosystems). The sequences were assembled using Genious 10.2.3. The nearly complete 16S rRNA gene sequence obtained was used for a similarity search in EzTaxon-e program (http://www.ezbiocloud.net/) for the valid taxonomic names. The bacterial species that closely resembled the query sequences were then used for alignment and phylogenetic analysis in MEGAX software (*Kumar et al., 2018*). Initially, the sequences were aligned using MUSCLE (*Edgar, 2004*) and the Neighbor Joining method (*Saitou & Nei, 1987*) was used to reconstruct the phylogenetic tree employing Kimura 2-parameter model (*Kimura, 1980*) with 1000 bootstraps. Phylogenetic trees were also constructed using maximum-likelihood (*Felsenstein, 1981*)and minimum evolution methods (*Rzhetsky & M, 1992*). *Clostridium butyricum* ATCC 19398[T] was used as an out-group.

## Genomic features and comparison

For the whole genome sequencing of SG502[T], we used 0.3ng of the genomic DNA for library preparation. Library was sequenced on an Illumina MiSeq using 2x 250 paired-end V2 chemistry. Genome was assembled from raw fastq files using Unicycler which builds an initial assembly graph from short reads using the de novo assembler SPAdes3.11.1 (*Bankevich et al., 2012*). Quality assessment for the assemblies was performed using QUAST (*Gurevich et al., 2013*). Genome annotation was performed using Prokka 1.13 (*Seemann, 2014*). The genome of SG502[T] was visualized using DNAplotter (*Carver et al., 2009*).

We compared the genome of SG502[T] to that of 16S-phylogeny of closely related species *C. innocuum* DSM 1286[T], *Longicatena caecimuris* DSM 29481[T], *Eubacterium dolichum* DSM 3991[T], *Faecalicoccus pleomorphus* DSM 20574[T], *Faecalitalea cylindroides* ATCC 27803[T], *Holdemanella biformis* DSM 3989[T] and *Dielma fastidiosa* DSM 26099[T]. In addition, we compared SG502[T] with the proposed members of the genus *Erysipelatoclostridium*, *C. clocleatum* DSM 1551[T], *C. ramosum* DSM 1402[T], *C. saccharogumia* DSM 17460[T] and *C. spiroforme* DSM 1552[T]. We used the average nucleotide identity (OrthoANI) (*Lee et al., 2016*) and digital DDH (*Meier-Kolthoff & Goker, 2019*) for calculating phylogenomic similarity. We also performed average amino acid identity (AAI) using AAI calculator (*Konstantinidis & Tiedje, 2005*) to determine protein level genome differences.

## RESULTS

The SG502[T] strain was isolated from the healthy human fecal sample during the culturomics study of the human gut microbiota. The colonies of the strain appeared white, smooth and convex with entire edges. The cells were initially subjected to MALDI-ToF MS (Fig. 1A) which revealed the score <1.70 suggesting no identification. Thus, further phenotypic characterization and genetic based methods were employed for identification of the strain.

Morphologically, individual cells of the strain appeared to be gram-positive rods. The cell was observed to be slender with tapering ends with $1.5 \times 0.35\,\mu$m in dimensions (Fig. 1B and Table 1) under SEM. No flagella were observed under SEM suggesting its non-motile nature which was also validated by TTC assay. The strain also lacked endospores, similar to what has been previously reported for the members of *Erysipelotrichaceae* (*Verbarg et al., 2004*). The strain grew in a pH range of 6.0–7.5 with optimal growth at pH 7.0. It could grow anaerobically over the temperature range of 25–45 °C with optimal growth at 37 °C.

A

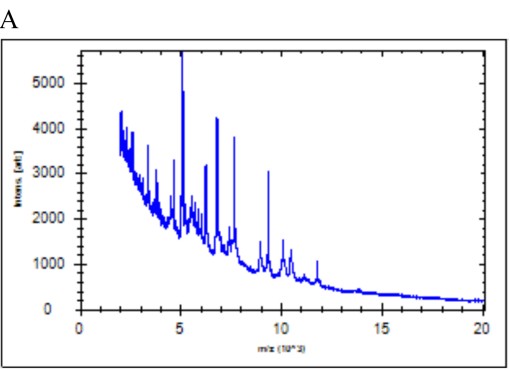

B

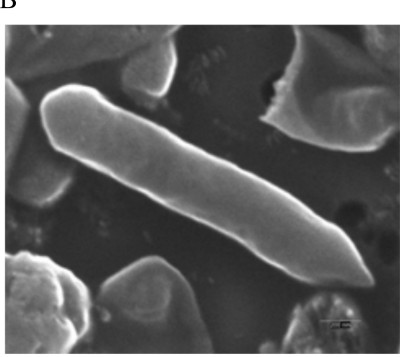

**Figure 1** **(A) MALDI-ToF reference spectrum obtained for SG502$^T$. (B) Scanning electron micrograph of strain SG502$^T$.** Cells were imaged after culturing in anaerobic conditions for 24 hours at 37 °C in BHI-M medium. Bar, 200 nm.

The strain grew well in BHI-M under anaerobic conditions but under aerobic conditions, the growth was comparatively lower and slow confirming that the strain was a facultative anaerobe. Based on the results obtained from a carbon source utilization test (Biolog AN plate), the strain utilizes glucose, sorbitol, maltose, arbutin, D-fructose, L-fucose, palatinose, dextrin, turanose, D-trehalose, L-rhamnose, uridine, pyruvic acid methyl ester, pyruvic acid, 3-methyl-D-glucose, gentiobiose, maltotriose, ducitol, L-phenylalanine, $\alpha$-ketovaleric acid, N-acetyl-D-glucosamine, N-acetyl-$\beta$-D-mannosamine, cellobiose, $\alpha$-ketobutyric acid, D-galacturonic acid and N-acetyl-D-glucosamine. Also, SG502$^T$ assimilated sorbitol and maltose which were not utilized by its closest neighbor *C. innocuum* ATCC 14501$^T$. Furthermore, SG502$^T$ was unable to utilize sucrose, salicin, mannitol, lactose, and raffinose when compared to *C. innocuum*. Positive enzymatic activities for leucine arylamidase, cystine arylamidase, $\alpha$-chymotripsin and acid phosphates were observed for *C. innocuum* differentiating it from SG502$^T$. Detailed phenotypic and biochemical characteristics of the strain are presented in Table 1. Also, the major fatty acids content identified were C$_{18:1}$ $\omega$9c (29.82%), C$_{18:0}$ (22.55%) and C16:0 (14.7%) compared to *C. innocuum* ATCC 14501$^T$ with C$_{18:1}$ $\omega$9c (14.64%), C$_{18:0}$ (10.56%) and C16:0 (23.7%) (Table 2). The detailed comparison of the fatty acids in SG502$^T$ along with *C. innocuum* 14501$^T$ and *E. dolichum* DSM 3991$^T$ is given in Table 2. Additionally, the major SCFAs metabolite identified for SG502$^T$ was acetate in BHI-M medium. Low but detectable amounts of propionate and butyrate were produced by the strain SG502$^T$. The utilization of such broad substrates and production of SCFAs can be ecologically effective trait against pathogen colonization in the gut.

As the strain was not identified using MALDI-ToF, 16S rRNA sequence was amplified to obtain a continuous stretch of 1338 bp gene which was searched against the Eztaxon 16S rRNA gene database for identification. The closest species identified were all from the *Erysipelotrichaceae* family that included *C. innocuum* ATCC 14501$^T$, *L. caecimuris* strain PG-426-CC-2, *E. dolichum* DSM 3991$^T$ and *E. tortuosum* ATCC 25548$^T$ with 95.28%, 94.17%, 93.25%, and 92.75% sequence identities respectively. Currently, the cut off for the species and genus level classification of the bacteria based on 16S rRNA gene is <98.7% (*E*

Table 1 Differential phenotypic features of the strain SG502$^T$ and its closest phylogenetic neighbor *C. innocuum* ATCC 14501$^T$ and *E. dolichum* DSM 3991$^T$ identified using API ZYM (bioMerieux, France).

| Characteristics | SG502$^T$ | ATCC 14501$^T$ | DSM 3991$^T$ (□) |
|---|---|---|---|
| Cell shape | Spindle | Rods | Rods |
| Gram stain | + | + | + |
| Growth at 37 °C (an.) | + | + | + |
| Optimal pH | 7 | 7 | 7 |
| Motility | − | − | − |
| Size (μ) | $1.5 \times 0.35$ | $2.0 - 4.0 \times 0.4 - 1.0$ | $1.6 - 6.0 \times 0.4 - 0.6$ |
| **Carbon sources utilization** | | | |
| Glucose | + | + | + |
| Sucrose | − | + | − |
| Salicin | − | + | − |
| Mannitol | − | + | − |
| Lactose | − | − | − |
| Sorbitol | + | − | − |
| Maltose | + | ± | + |
| D-Trehalose | + | + | + |
| Raffinose | − | + | − |
| Cellobiose | + | + | − |
| **Enzyme activity (API ZYM)** | | | |
| Alkaline phosphatase | + | + | ND |
| Esterase (C4) | − | − | ND |
| Esterase Lipase (C8) | − | − | ND |
| Lipase (C14) | − | − | ND |
| Leucine arylamidase | − | + | ND |
| Valine arylamidase | − | − | ND |
| Cystine arylamidase | − | + | ND |
| Trypsin | − | − | ND |
| $\alpha$-chymotrypsin | − | + | ND |
| Acid phosphatase | − | + | ND |
| Naphthol-As-Bi-phosphopydrolase | − | − | ND |
| $\alpha$-galactosidase | − | − | ND |
| $\beta$-galactosidase | − | − | ND |
| $\beta$-glucuronidase | − | − | ND |
| $\alpha$-glucosidase | − | − | ND |
| $\beta$-glucosidase | − | − | ND |
| N-acetyl- $\beta$-glucosaminidase | − | − | ND |
| $\alpha$-mannosidase | − | − | ND |
| GC content (%) | 32.88 | 44.5 | 39 |

Notes.
□Data obtained from *Moore, Johnson & Holdeman (1976)*.
ND, not determined.

**Table 2  Cellular fatty acid contents percentages (%) of strain SG502$^T$ compared to its phylogenetic neighbors *C. innocuum* ATCC 14501$^T$ and *E. dolichum* DSM 3991$^T$.** Those fatty acids which were not separated using MIDI system were considered as summed features. Summed feature 5 contains C$_{15:0}$ DMA or C$_{14:0}$ 3-OH; summed feature 8 contains C$_{17:1cis9}$ or C$_{17:2}$ and summed feature contains C$_{18:1}$c11/t9/t6 or UN17.83Q.

| Characteristics | SG502$^T$ | ATCC 14501$^T$ | DSM 3991$^T$ ([a]) |
|---|---|---|---|
| **Straight chain** | | | |
| C$_{10:0}$ | 0.31 | 0.32 | 1 |
| C$_{12:0}$ | 1.74 | 3.28 | – |
| C$_{14:0}$ | 3.54 | 8.85 | 1.6 |
| C$_{16:0}$ | 14.7 | 23.7 | 22.8 |
| C$_{16:0}$ aldehyde | 0.42 | 2.78 | – |
| C$_{17:0}$ | 1.41 | – | 1.8 |
| C$_{18:0}$ | 22.55 | 10.56 | 17.1 |
| **Dimethyl acetal (DMA)** | | | |
| C$_{16:0}$ DMA | 1.46 | 9.95 | – |
| C$_{18:0}$ DMA | 3.62 | 2.83 | – |
| **Unsaturated** | | | |
| C$_{16:1}$ $\omega$9c | 0.99 | 6.51 | – |
| C$_{16:1}$ $\omega$7c | 6.47 | 10.59 | – |
| C$_{18:1}$ $\omega$9c | 29.82 | 14.64 | 33.9 |
| C$_{18:1}$ $\omega$7c | 10.77 | 3.08 | - |
| Summed Feature 8 | 0.54 | – | – |
| Summed Feature 10 | 10.77 | 3.08 | – |

**Notes.**

[a] Data from *Paek et. al (2017)*.

*& J, 2006*) and <94.5% identity (*Yarza et al., 2014*) respectively. Thus, the strain SG502$^T$ and *C. innocuum* were suggested to fall within same genus but different species. The phylogenetic analysis also revealed that the isolate belonged to *Erysipelotrichaceae* family where the strain SG502$^T$ was closely associated to *C. innoccum* ATCC 14501$^T$ but further from *L. caecimuris* strain PG-426-CC-2, *E. dolichum* DSM 3991$^T$ and *E. tortuosum* ATCC 25548$^T$ which altogether formed a larger clade (Fig. 2). The separation of these four species from the strain SG502$^T$ did not depend on the phylogenetic algorithm and was supported by an 100% bootstrap value. To further differentiate the strain, we sequenced the whole genome of the strain and is visualized in Fig. 3. The draft genome of the strain SG502$^T$ was 2,683,792 bp with 32.88 mol% G+C content. The largest contig was of 154,144 bp and N$_{50}$ was 52,214. The total number of predicted coding sequences, tRNAs, rRNAs, and tmRNAs was 2654, 49, 2 and 1 respectively.

*C. innoccum* was the nearest neighbor of SG502$^T$ based on 16S rRNA phylogeny. *C. innoccum* along with *C. cocleatum, C. saccharogumia, C. ramosum,* and *C. spiroforme* were suggested to be reclassified previously into genus *Erysipelatoclostridium* with *C. innocuum* needing further reclassification (*Yutin & Galperin, 2013*). Therefore, we checked for the 16S identity of SG502$^T$ with the other members of this proposed genera *C. cocleatum, C. saccharogumia, C. ramosum,* and *C. spiroforme* in NCBI. These species were found to
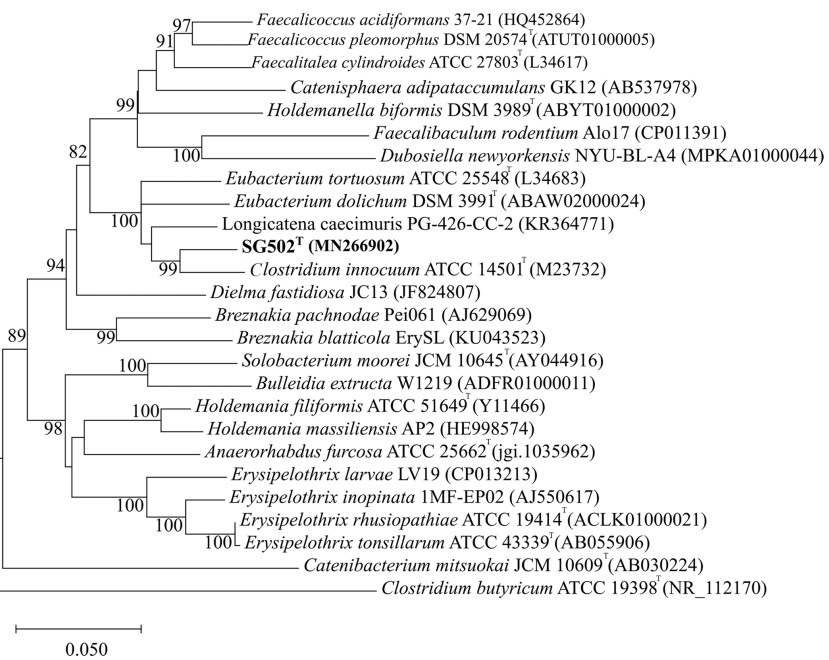

**Figure 2** **Neighbor Joining tree of 16S rRNA gene sequences of SG502^T with related species under** *Erysipelotrichaceae* **family.** GenBank accession numbers of the 16S rRNA gene sequences are given in parentheses. The sequences were aligned using MUSCLE (*Edgar, 2004*) and the evolutionary distances were computed using Kimura 2-parameter method to obtain the phylogenetic tree in MEGAX (*Kumar et al. 2018*) after 1,000 bootstrap tests (shown as percentages with associated taxa clustered together next to the branches. The tree is drawn to scale, with branch lengths measured in the number of substitutions per site. Bar, 0.05 substitutions per nucleotide position. *Clostridium butyricum* ATCC 19398^T was used as an outgroup.

be 84.45%, 84.49%, 85.10% and 85.014% identical respectively which demonstrated that SG502$^T$ should not be placed into same genera with these species. We also compared the genomic properties of the strain with its 16S rRNA based phylogenetic neighbors along with *C. innocuum* and the members of formerly proposed genus *Erysipelatoclostridium, C. cocleatum, C. saccharogumia*, *C. ramosum*, and *C. spiroforme*. The genomic sizes and G+C content of the members of the neighbors of the strain were found to vary as shown in Table 3. *C. innocuum* was 4,772,018 bp in length with 43.4 mol% G+C content, while for SG502$^T$, genome length was 2, 683,792 bp with 32.88 mol% G+C content. The genome sizes and G+C content of the neighboring species were highly variable compared to SG502$^T$ (Table 3). Because of such high differences in the genome properties of SG502$^T$, we performed further comparison for classifying SG502$^T$ as a novel genus. Hence, the genome of the strain was compared to its neighbors using OrthoANI as shown in Fig. 4. The strain SG502$^T$ was 71.8% similar to its nearest neighbor *C. innocuum* and had lower similarities with other neighbors. (Figs. 4A, 4B). The proposed cut-off for OrthoANI for a new species is 95–96% (*Kim et al., 2014*; *Lee et al., 2016*). The dDDH was only 19.8% between SG502$^T$ and *C. innocuum* (Table 4). One of the major methods to demarcate the genus is to calculate average amino acid identity (AAI) between the genomes with the
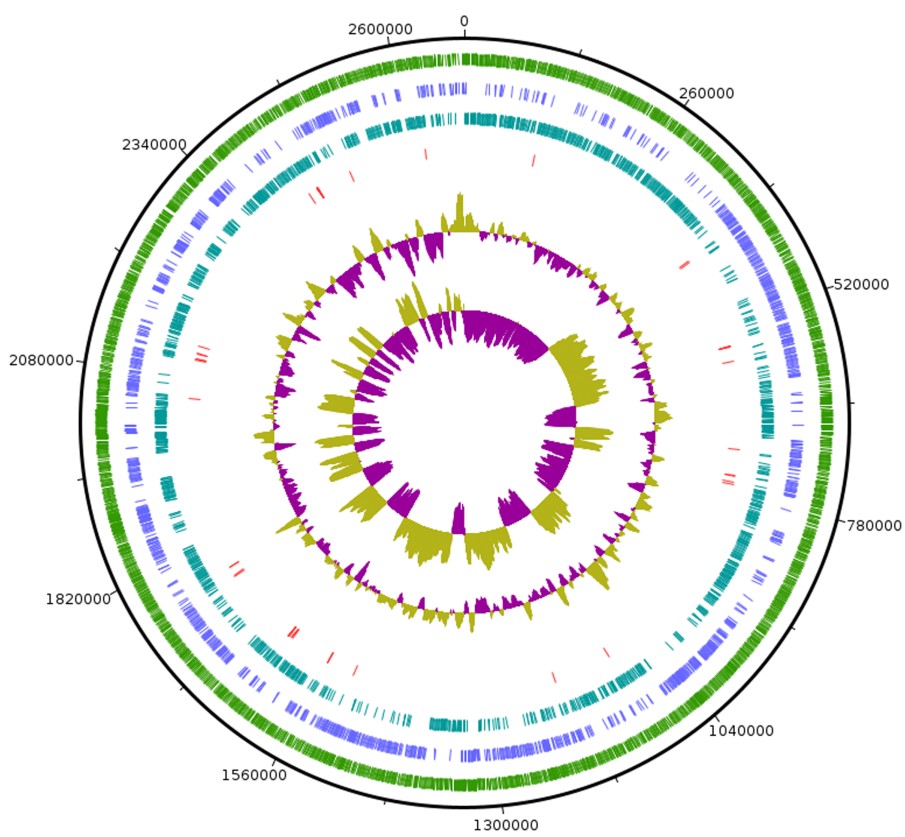

**Figure 3  Circular visualization of genome of SG502^T.** From outside to inside, "green" circle represents total number of CDS, "blue" represents number of CDS in positive strand, "bluish green" represents number of CDS in negative strand and "red" represents tRNAs position in the genome. The innermost circle represents GC skewness and circle inside to tRNAs represent average GC content. "Magenta" color represents GC at lower level while "Olive green" color represents GC at higher level.

possibility of novel genus if the AAI values are in the range of 65–72% (*Konstantinidis & Tiedje, 2007*). The strain SG502^T showed highest AAI with *C. innocuum* (69.65%) followed by *L. caecimuris* (63.45%) and *E. dolichum* (63.02%) (Fig. 5) supporting the designation of strain SG502^T in a novel genus.

## DISCUSSION

Recently, next generation sequencing and high-throughput culturing methods has been employed for large scale culture of the unknown gut microbiota. This new approach termed as "culturomics" has evolved as a tool to culture previously uncultured bacteria (*Browne et al., 2016*; *Lagier et al., 2016*). However, such culture independent studies have also highlighted that the diverse population of gut bacteria are yet to be cultivated (*Almeida et al., 2019*; *Lagier et al., 2012*). The pure culture of the bacteria is essential to elucidate the role of these organisms in health and diseases for both experimental model and therapeutics purposes (*Daillere et al., 2016*; *Kobyliak et al., 2016*; *Vetizou et al., 2015*). In this study, we report the culturing and characterization of a previously uncultured bacterium SG502^T
**Table 3  Genomic comparison of the strain SG502$^T$ with its neighbors.**

| Bacteria | Contigs | size (bp) | G+C (%) | rRNA | tRNA | tmRNAs | CDSs | Accession |
|---|---|---|---|---|---|---|---|---|
| SG502$^T$ | 119 | 2,683,792 | 32.9 | 2 | 49 | 1 | 2,654 | |
| *Clostridium innocuum* DSM 1286$^T$ | 18 | 4,772,018 | 43.4 | 13 | 48 | 1 | 4,653 | AGYV00000000 |
| *Longicatena caecimuris* DSM 29481$^T$ | 45 | 2,945,084 | 37.8 | 1 | 43 | 1 | 2,708 | SMBP00000000 |
| *Eubacterium dolichum* DSM 3991$^T$ | 25 | 2,190,453 | 38.1 | 6 | 41 | 1 | 2,144 | ABAW00000000 |
| *Faecalicoccus pleomorphus* DSM 20574$^T$ | 47 | 1,992,636 | 39 | 5 | 44 | 0 | 1,961 | ATUT00000000 |
| *Faecalitalea cylindroides* ATCC 27803$^T$ | 143 | 1,944,726 | 34.7 | 0 | 23 | 1 | 1,904 | AWVI00000000 |
| *Holdemanella biformis* DSM 3989$^T$ | 161 | 2,415,920 | 33.8 | 4 | 45 | 0 | 2,347 | ABYT00000000 |
| *Dielma fastidiosa* DSM 26099$^T$ | 145 | 3,575,363 | 40 | 3 | 50 | 1 | 3,447 | CAEN00000000 |
| *Clostridium cocleatum* DSM 1551$^T$ | 88 | 2,957,106 | 28.5 | 8 | 42 | 1 | 2,670 | FOIN00000000 |
| *Clostridium saccharogumia* DSM 17460$^T$ | 135 | 3,141,523 | 30.2 | 5 | 56 | 1 | 2,751 | JMLH01000000 |
| *Clostridium spiroforme* DSM 1552$^T$ | 16 | 2,507,485 | 28.6 | 13 | 58 | 1 | 2,330 | ABIK00000000 |
| *Clostridium ramosum* DSM 1402$^T$ | 16 | 3,234,795 | 31.4 | 9 | 41 | 1 | 3,056 | ABFX00000000 |

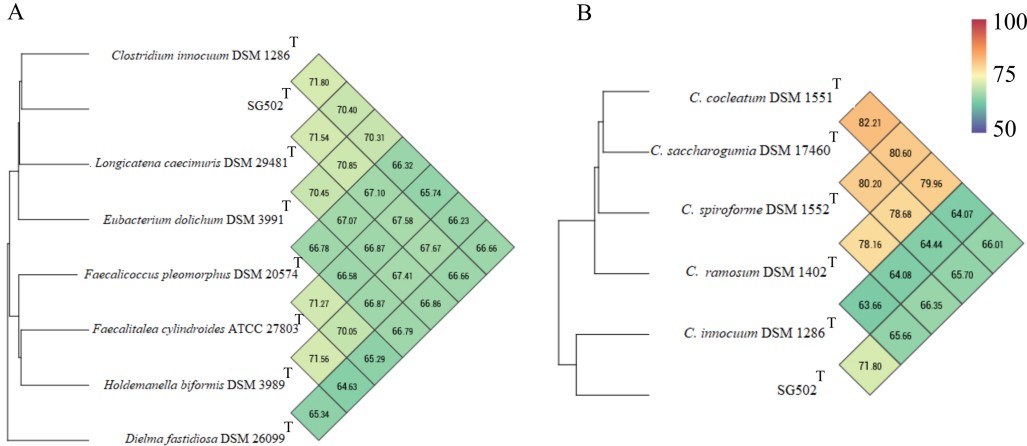

**Figure 4  Genomic comparison of SG502$^T$ genome with its neighbors using OrthoANI in OAT software.** (A) Comparison with 16S phylogenetic neighbors (B) Comparison with formerly proposed members of genus *Erysipelatoclostridium*. Color scale indicates % identity between the genomes.

from the healthy human fecal samples that belongs to a new genus and species. Also, we employed taxono-genomics approach (*Fournier & Drancourt, 2015*) to determine the phenotypic and genetic properties of the taxon.

16S rRNA based gene sequence homology is the widely used method to determine the novelty of the prokaryotic organism with varying threshold values at distinct taxonomic

**Table 4  Genomic comparison of the strain SG502$^T$ with its neighbors 16S rRNA phylogenetic neighbors and proposed *Erysipelatoclostridium* members using TYGS.**

| Bacteria | dDDH (d4, %) | C.I. (d4, %) | G+C difference (%) |
|---|---|---|---|
| *Clostridium innocuum* DSM 1286$^T$ | 19.8 | [17.6–22.2] | 10.48 |
| *Longicatena caecimuris* DSM 29481$^T$ | 20.5 | [18.3–22.9] | 4.86 |
| *Eubacterium dolichum* DSM 3991$^T$ | 20.6 | [18.4–23.0] | 5.22 |
| *Faecalicoccus pleomorphus* DSM 20574$^T$ | 19.6 | [17.4–22.0] | 6.14 |
| *Faecalitalea cylindroides* ATCC 27803$^T$ | 18.9 | [16.7–21.3] | 1.79 |
| *Holdemanella biformis* DSM 3989$^T$ | 19 | [16.8–21.4] | 0.9 |
| *Dielma fastidiosa* DSM 26099$^T$ | 20.1 | [17.9–22.5] | 7.08 |
| *Clostridium cocleatum* DSM 1551$^T$ | 23.5 | [21.2–26.0] | 4.37 |
| *Clostridium saccharogumia* DSM 17460$^T$ | 18.6 | [16.4–21.0] | 2.72 |
| *Clostridium spiroforme* DSM 1552$^T$ | 19.8 | [17.6–22.2] | 4.31 |
| *Clostridium ramosum* DSM 1402$^T$ | 22.4 | [20.1–24.8] | 1.5 |

levels (*Clarridge 3rd, 2004*; *Kim et al., 2014*) . Therefore, we performed the 16SrRNA based phylogenetic analysis of the strain SG502$^T$ which showed it as a member of *Erysipelotrichaceae* family. Under this family, it clustered together with *Clostridium innoccum*, *Longicatena caecimuris*, *Eubacterium dolichum* and *Eubacterium tortuosum* with *C. innocuum* as a closest member. *C. innocuum* along with other members of misclassified *Clostridia* under *Erysipelotrichaceae* family were proposed to be reclassified into gen. nov. *Erysipeloclostridium*. The members of this proposed genus *Erysipelatoclostridium* are gram positive, nonmotile, obligately anaerobic straight or helically curved rods which rarely forms spores. The G+C content is lower and varies from 27-33 mol% (*Yutin & Galperin, 2013*). However, *C. innocuum* was identified to be a distantly related member of *Erysipelatoclostridium* with higher G+C content of 43–44% with need of reclassification (*Yutin & Galperin, 2013*). In this context, we also searched for the 16S based identity of the strain SG502$^T$ with the proposed members of genus *Erysipelatoclostridium*. Nevertheless, the proposed members of genus *Erysipelatoclostridium* were <86% similar at 16S sequence level, suggesting the uniqueness of SG502$^T$.

Phenotypically, SG502$^T$ revealed several differences in carbon sources utilization, enzymatic activity and fatty acid when compared to its phylogenetic neighbors (Tables 1 and 2). In addition, whole genome sequence comparison revealed its distinctiveness with respect to 16S phylogenetic members and the proposed members of *Erysipelatoclostridium* genus (Tables 3 and 4). Furthermore, OrthoANI based genomic comparison with 16S phylogenetic neighbors showed as high as 71.54% similarity with *L. caecimuris* DSM 29481$^T$. Also, the genome of SG502$^T$ was only 82.21% similar with *C. cocleatum* DSM 1551$^T$ which is a member of *Erysipelatoclostridium* genus (Fig. 4). Major differences were evident in dDDH and amino acid composition comparison as well (Table 4, Fig. 5). Finally, the genome size of the nearest neighbor *C. innocuum* was nearly twice that of SG502$^T$ and the difference of G+C content was comparatively high (>10 mol%) suggesting

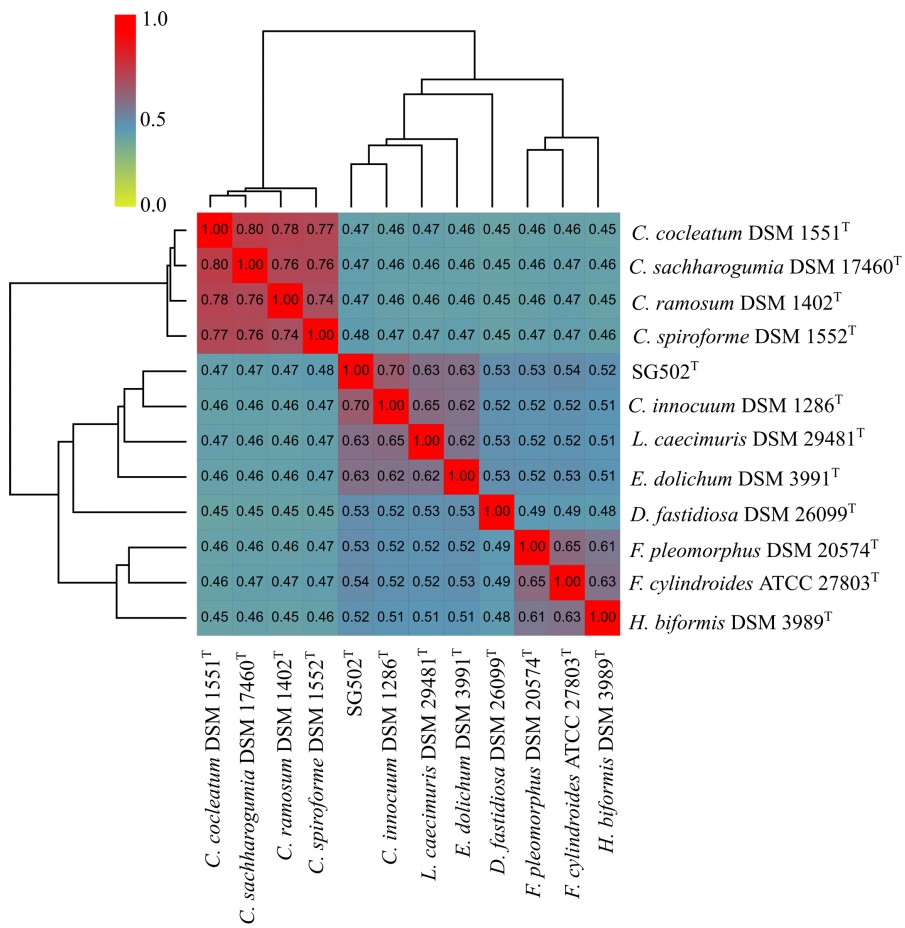

**Figure 5** Average Amino acid composition comparison of the strain SG502<sup>T</sup> with its neighbors 16S
rRNA phylogenetic neighbors and proposed *Erysipelatoclostridium* members using AAI calculator.

that the strain is not close to *C. innocuum* genetically which means that SG502$^T$ require the
placement in a separate genus.

## CONCLUSION

Despite 95.15% 16S rRNA similarity of SG502$^T$ with its nearest neighbor *C. innocuum*,
the differences in its physiological, biochemical, and whole genome sequence suggest
its placement in a novel genus. Hence, we propose creation of a novel genus *Dakotella*
under family *Erysipelotrichaceae* and classification of SG502$^T$ under new genus *Dakotella*
as *Dakotella fusiforme* SG502$^T$.

## DESCRIPTION OF *DAKOTELLA* GEN. NOV.

*Dakotella* (*Da.ko.tel'la*. M.L. dimin. ending -*ella*; N.L. fem. n. *Dakotella*, from the place of
isolation, State of South Dakota, USA).

The type strain is elongated spindle shaped. The closest phylogenetic neighbor is *C. innocuum* ATCC 14501[T] with corresponding dDDH of 19.8%. The relative genomic G+C difference with *C. innocuum* is 10.48 mol %. The OrthoANI of the isolate and the type species *C. innocuum* is 71.8%. *E. dolichum* DSM 3991[T] and *Longicatena caecimuris* DSM 20574[T] are distantly related with OrthoANI of 70.85% and 71.84% and dDDH of only 20.6% and 20.5% respectively. The corresponding difference in the G+C content is 5.22% and 4.86% with *E. dolichum* DSM 3991[T] and *Longicatena caecimuris* DSM 20574[T] respectively. Such differences support the creation of novel genus to accommodate SG502[T]. The G+C content of the genomic DNA of the type strain is 32.88 mol%. The type species is *Dakotella fusiforme*.

## DESCRIPTION OF *DAKOTELLA FUSIFORME* SP. NOV. SG502[T]

*Dakotella fusiforme* (fu.si.for'me. L. masc. *fusus* spindle): referring to shape

The cells of the bacterium are anaerobic, gram-positive non-motile rods. The average size of the cell is 1.5×0.35 μm. Bacterial colonies on BHI-M agar are white, convex and entire approximately 0.1 cm in diameter. The optimum temperature and pH for the anaerobic growth are 37 °C and 7.0 respectively. The strain SG502[T] utilizes glucose, sorbitol, maltose, arbutin, D-fructose, L-fucose, palatinose, dextrin, turanose, D-trehalose, L-rhamnose, uridine, pyruvic acid methyl ester, pyruvic acid, 3-methyl-D-glucose, gentiobiose, maltotriose, ducitol, L-phenylalanine, a-ketovaleric acid, N-acetyl-D-glucosamine, N-acetyl-b-D-mannosamine, cellobiose, a-ketobutyric acid, D-galacturonic acid and N-acetyl-D-glucosamine. Positive enzymatic reactions were observed for alkaline phosphatase only. The primary short-chain fatty acid produced by the strain is acetate while small amounts of propionate and butyrate were also noted. The major cellular fatty acids of the strain SG502[T] are $C_{18:1}$ $\omega$9c, $C_{18:0}$ and $C_{16:0}$.. The type strain, SG502[T] (=DSM 107282[T]=CCOS 1889[T]), was isolated from a healthy human fecal sample. The genomic size of the strain is 2, 683,792 bp and G+C content of the strain SG502[T] is 32.88 mol%.

## PROTOLOGUE

The GenBank accession number for the 16S rRNA gene sequence of the strain SG502[T] is MN266902. The GenBank BioProject ID number for the draft genome sequence of the strain SG502[T] is PRJNA494608 .

**Abbreviations**

| MALDI-ToF | Matrix Assisted Laser Desorption/Ionization-Time of Flight |
|---|---|
| ANI | Average Nucleotide Identity |

## ACKNOWLEDGEMENTS

The authors would like to thank Electron Microscopy Core Facility at the Bowling Green State University, Ohio, USA for assistance with scanning electron microscopy.

### Funding

This work was supported by the USDA National Institute of Food and Agriculture, Hatch projects SD00H532-14 and SD00R540-15, and a grant from the South Dakota Governor's Office of Economic Development awarded to Joy Scaria. The funders had no role in study design, data collection and analysis, decision to publish, or preparation of the manuscript.

### Grant Disclosures

The following grant information was disclosed by the authors:
The USDA National Institute of Food and Agriculture, Hatch projects: SD00H532-14, SD00R540-15.
South Dakota Governor's Office of Economic Development.

### Competing Interests

The authors declare there are no competing interests.

### Author Contributions

- Sudeep Ghimire performed the experiments, analyzed the data, prepared figures and/or tables, authored or reviewed drafts of the paper, and approved the final draft.
- Supapit Wongkuna performed the experiments, authored or reviewed drafts of the paper, and approved the final draft.
- Joy Scaria conceived and designed the experiments, authored or reviewed drafts of the paper, and approved the final draft.

### Human Ethics

The following information was supplied relating to ethical approvals (i.e., approving body and any reference numbers):
    The South Dakota State University Institutional Review Board granted ethical approval to carry out the study (IRB approval number IRB-1512008-EXP).

### Data Availability

    The draft genome sequence of the strain SG502T is available at GenBank BioProject: PRJNA494608.

### New Species Registration

The following information was supplied regarding the registration of a newly described species:
    https://www.ncbi.nlm.nih.gov/Taxonomy/Browser/wwwtax.cgi?id=1505663.

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
