# Peer review of "Description of a new member of the family Erysipelotrichaceae: Dakotella fusiforme gen. nov., sp. nov., isolated from healthy human feces"

_PeerJ, doi:10.7717/peerj.10071_

## Round 0.1 · original submission · Major Revisions

Thank-you for your manuscript submission to PeerJ. As indicated above, a number of changes are required before it can be considered for publication. Please respond to all of the suggestions provided by the reviewers, providing point-to-point feedback describing your updates. In particular, please respond to the concern about the indicated genus name and whether or not it is officially valid. Also, as indicated by reviewer 1, a broader discussion of the potential importance of this organism as part of the gut microbial population is warranted. Simply indicating that these organisms are "associated with host metabolism and inflammatory diseases" is insufficient. Finally, as noted by reviewer 1, the genomic sequencing data is barely discussed, and does not contribute to the knowledge or understanding of this organism as described by the manuscript. The manuscript would be much stronger if the genomic sequencing data was more complete, and was a focal-point of the results and discussion.

Reviewer 1 ·

Basic reporting

This manuscript was written in clear English and the structure of the article conforms to an acceptable format of PeerJ. However, sufficient Background/Introduction was not provided and was only within reach of some specialists. The authors need to make a significant revision of Introduction for broad readers.

Experimental design

see below

Validity of the findings

see below

Additional comments

Ghimire et al. described the isolation and physiological characterization of a new member of the genus Erysipelatoclostridium species isolated from human feces, and they proposed it as a novel species Erysipelatoclostridium fusiforme sp. nov within the family Erysipelotrichaceae. This is a fairly straightforward paper describing a novel species in the genera Erysipelatoclostridium within the family Erysipelotrichaceae.
Overall, experiments related to the phenotypic and chemotaxonomic analyses are well designed and well conducted. Nevertheless, there are some critical concerns that must be addressed. Some major comments are shown as follows.

1. This manuscript is too descriptive to understand the importance of this study for broad readers, though the authors performed phenotypic and chemotaxonomic characterization according to a conventional manner of International Journal of Systematic and Evolutionary Microbiology. The authors should add effective discussion about physiology and ecological role of the isolated microorganism in human gut.

2. The authors rarely discuss about the result of genome sequence of the isolated strain. Moreover, the quality of the genome sequence is low (i.e., 146 contigs and 2.4Mbp), while the relative strain (Erysipelatoclostridium innocuum) consists of 4.8Mbp genome. Therefore, the authors should conduct additional genome sequencing and comparative genome analyses.

3. Cell morphology of strain SG502T is still unclear, since Figure 1 includes some different-shaped cells other than a rod-shaped cell. Cell morphology is one of the most important indicators of taxonomy. Therefore, another micrograph such as transmission electron micrograph and phase contrast micrograph are highly required. Does this isolate form spores?

4. The authors should submit additional figure of MALDI-ToF and discuss about the result.

Reviewer 2 ·

Basic reporting

The genus name Erysipelatoclostridium has not been listed in the Approved Lists of Bacterial Names (Bergey’s Manual). (LPSN bacterio. net)
It is required to describe the novel species with a proposal of a novel genus including closely related species properly.

Expressions for the taxonomical terms should be improved.

Experimental design

The novel species is proposed based on the unvalidated genus name. Thus, the authors should check the valid names of the closely related species and consider how to propose the novel species (or genus) name validly.

Validity of the findings

Basic characterization of the novel strains was carried out almost properly.

Additional comments

The genus name Erysipelatoclostridium has not been listed in the Approved Lists of Bacterial Names (Bergey’s Manual). (LPSN bacterio. net)
It is required to describe the novel species with a proposal of a novel genus including closely related species properly.
Expressions for the taxonomical terms should be improved.

Minor comments:
Strain name: Delete ‘the’ from all strain names (i.e., strain SG502T, strains SG502T and ATCC 14501T).
(You can use the term ‘the strain’ without the strain number.)

Add the definite article ‘the’ to express taxonomic ranks (e.g., within the family Clostridiaceae, of the genus Clostridium, to the genera Clostridium and Eubacterium).

line 150, 200-201: Individual cells of strain SG502T appeared to be Gram-positive rod-shaped with 1.5×0.35 μm in
line 203: Utilizes glucose, sorbitol,
Figure 1: Add “T” to the numbers of all type strains in the figure. (e.g., ATCC 19398T, ‘T’ is a superscript.)

Table 1: Differential phenotypic features of strain SG502T. (The table includes characteristics other than ‘biochemical’ features.)
Size (μm)

Table 2: ‘contains C15:0 DMA or C14:0 3-OH; summed feature 8 contains
C17:1 cis 9 or C17:2 and summed feature contains C18:1 c11/t9/t6 or
UN17.83Q.’
Subscripts for 15:0, 14:0, 17:1, 17:2 and 18:1.

---

## Round 0.2 · accepted · Accept

Thank-you for performing additional sequencing and revising the first draft of your manuscript to address reviewer comments. I am now happy to accept your manuscript for publication.